# Recent Advances in Natural Product-Based Hybrids as Anti-Cancer Agents

**DOI:** 10.3390/molecules27196632

**Published:** 2022-10-06

**Authors:** Eleni Sflakidou, George Leonidis, Eirini Foroglou, Christos Siokatas, Vasiliki Sarli

**Affiliations:** Department of Chemistry, Aristotle University of Thessaloniki, University Campus, 54124 Thessaloniki, Greece

**Keywords:** natural product, hybrid, PROTAC, AUTOTAC, folate hybrid, antibody–drug conjugate, peptide–drug conjugate, aptamer–drug conjugate, steroidal hybrid

## Abstract

Cancer is one of the top leading causes of death worldwide. It is a heterogenous disease characterized by unregulated cell proliferation and invasiveness of abnormal cells. For the treatment of cancer, natural products have been widely used as a source of therapeutic ingredients since ancient times. Although natural compounds and their derivatives have demonstrated strong antitumor activity in many types of cancer, their poor pharmacokinetic properties, low cell selectivity, limited bioavailability and restricted efficacy against drug-resistant cancer cells hinder their wide clinical application. Conjugation of natural products with other bioactive molecules has given rise to a new field in drug discovery resulting to the development of novel, bifunctional and more potent drugs for cancer therapy to overcome the current drawbacks. This review discusses multiple categories of such bifunctional conjugates and highlights recent trends and advances in the development of natural product hybrids. Among them, ADCs, PDCs, ApDCs, PROTACs and AUTOTACs represent emerging therapeutic agents against cancer.

## 1. Introduction

Cancer is a heterogenous disease characterized by unregulated cell proliferation and invasiveness of abnormal cells. It is considered a genetically inherited disease in 5 to 10% of cases but the majority of cases are the result of exogenous factors, such as environmental stimuli, diet, lifestyle and pollution [1]. Herbs and natural products can be considered as traditional cancer treatment used since the dawn of civilization. Ancient Greeks, Arabs, Chinese and Indians were pioneers in the development of herbal medicines with multiple applications including cancer [2,3]. Naturally derived compounds like curcumin, quercetin and resveratrol are known for their superior antitumor activities, while paclitaxel and doxorubicin are FDA-approved drugs for cancer therapy [4]. Their structural complexity and unique properties render them able to serve specific biological functions and overcome problems that synthetic drugs face, like oral administration stability [5]. However, the wide application of natural compounds in medicine is hindered by challenges like their insufficient pharmacokinetics and pharmacodynamics, poor solubility, stability and limited bioavailability [6]. Moreover, their identification and sufficient isolation are extremely demanding and expensive, leading pharmaceutical companies to reduce investments on natural product-based drug discovery programs [7].

One of the growing areas in modern drug discovery focuses on the synthesis of bifunctional conjugates of natural products to overcome the drawbacks to produce novel and more efficient drugs for cancer treatment. Such natural product-based hybrids include small molecules, antibodies, peptides, aptamers and other natural products, as well as metal complexes, such as bioactive moieties (Figure 1) [8]. These types of conjugates improve natural products’ selectivity towards cancer cells, thus reducing their toxicity. Their antitumor efficacy is enhanced either through synergistic effect or improvement of their bioavailability [9]. In addition, these alternative approaches can evidently overcome multiple drug resistance by targeting different biological pathways in many types of tumors with promising results for clinical applications [10]. This review discusses multiple categories of such bifunctional hybrid compounds and highlights recent trends and advances in the development of natural product hybrids. Among them, ADCs, PDCs, ApDCs, PROTACs and AUTOTACs represent emerging therapeutic agents against cancer.

## 2. Hybrids of Natural Products with Small Molecules

The covalent conjugation of a natural product with a second bioactive small molecule may generate a new compound with superior properties compared to the parent natural product, such as improved selectivity, metabolism resistance and better pharmacodynamics and physicochemical characteristics [11]. Furthermore, mechanistic differentiation that can alter the mode of action by which a natural product exerts its antitumor activity may also occur. For example, small molecules can act as ligands that induce autophagy and proteolysis or activate apoptotic pathways. The conjugation of natural products with small molecules, such as heterocyclic compounds, phenyl analogues, other natural products, folic acid, steroids and ligands that induce proteolysis and autophagy, as well as their anticancer efficacy, will be discussed in this section.

### 2.1. Hybrids of Natural Products with Heterocyclic Compounds

Heterocyclic scaffolds are prevalent in many FDA-approved pharmaceutical compounds like 5-flouorouracil, methotrexate and doxorubicin [12]. These moieties are considered as crucial parts in the development of novel drugs; therefore, a plethora of natural products conjugated with heterocyclic compounds have emerged through the years. Ndreshkjana et al. reported a covalent conjugation of the secondary metabolite thymoquinone (TQ) with the conventional chemotherapeutic 5-fluorouracil (5-FU) (Figure 2) [13]. The efficiency of 5-FU in cancer therapy is significantly limited due to the development of drug resistance. The authors addressed drug resistance presented in colorectal cancer, caused by the accumulation of cancer stem cells. 5-FU alone failed to kill the stem cells, thus leading to relapse of the disease. The efficiency of hybrid and combinational treatments was examined along with individual TQ and 5-FU treatments against HCT116, HCEC and HT29 cell lines and CAM xenografts. Both combinational and hybrid treatments showed increased cytotoxicity in vitro by inhibiting the Wnt/β-Catenin and PI3K/Akt pathways, which play an indispensable role in the regulation of stem cells [14]. Hybrid treatment was more successful than combinational treatment against xenografted CAM as evidenced by diminished tumor growth. These results were attributed to the TQ-5-FU conjugate’s ability to selectively target the Wnt pathway in comparison to the combinational treatment. Additionally, TQ-5-FU hybrid (**1**) demonstrated a 4-fold higher downregulation of the PI3K/Akt pathway-associated genes compared to combinational treatment. This work highlighted the synergistic effect of TQ and 5-FU and the improved hybrid’s targeting ability derived from conjugation.

Chalconoids are α,β-unsaturated ketones synthesized by plants of the genus *Tephrosia*, *Fissistigma bracteolatum* as secondary metabolites [15]. Chalcones were proved to be very efficient against various cancer cell lines, while some analogues can activate programmed cell death via the caspase-dependent intrinsic mitochondrial pathway [16]. Wang et al. demonstrated the development of a novel chalcone and 1,2,3-triazole hybrid **2** and evaluated its antiproliferative activity against liver cancer cells (Figure 2) [17]. Triazole analogues are widely used in medicinal chemistry since they can enhance the antitumor activity of cytotoxic compounds and alter their pharmacokinetic properties [18,19]. The chalcone hybrid demonstrated low micromolar IC_50_ values against a series of cancer cell lines, compared to the control treatment. It should be highlighted that the newly developed chalcone hybrid exhibited potent antitumor activity in HepG2 xenografted mice compared to the control treatment (tumor weights = 1.3 g compared to 0.5 g, inhibitory rate: 60%.

Coumarins interacting with various cellular components are used as anticoagulants, antibiotics, antiproliferative and anti-HIV drugs. Dihydro-triazole rings were also utilized for the synthesis of coumarin-based hybrids (**3**, **4**) by Güner and his team to improve anticancer efficacy and overcome cisplatin resistance in tumors (Figure 2) [20]. The hybrids were evaluated alone against breast, cervical and lung cancer cells and normal human HEK293 cells. All hybrids except for the benzyl- and phenyl-substituted analogues were cytotoxic against both cancer and normal cells. Thus, phenyl- and benzyl-substituted hybrids were further studied due to their selectivity towards cancer cells. Combinational treatment with cisplatin in A549 cancer cell line revealed a better antiproliferative profile compared to cisplatin and conjugates alone. The chorioallantoic membrane model on fertilized hen eggs was applied for the assessment of the anti-angiogenic activity of the novel compounds. Combinational treatment demonstrated slightly better results than cisplatin or conjugates alone. It is currently believed that upregulation of ROS levels promotes cisplatin resistance in cancer cells [21]. Reactive oxygen species (ROS) are essential for vital cell functions, like proliferation and differentiation. Güner et al. managed to address this issue by demonstrating the suppression of antioxidant levels in lung cancer cells by their successful combinational treatment.

Apigenin is a natural flavone isolated from plant sources like parsley, celery, and chamomile [22]. Piperazine is a common element of a wide array of natural products and pharmaceutical substances. Long et al. extensively studied the properties of a novel apigenin-piperazine hybrid (**5**), according to QSAR studies (Figure 3), that exhibited low nanomolar inhibitory effect against PARP-1 [23] and moderate antitumor activity against A549 and SKOV-3 xenografted mice (50 mg/kg for 12 days, relative tumor volume = 0.74 compared to 2.12 of Olaparib, a PARP-1 inhibitor). Apigenin–piperazine hybrid was also evaluated for its ADME characteristics, pharmacokinetic parameters and healthy tissue toxicity with great results.

Quinolines are naturally occurring heterocyclic aromatic alkaloids that have been employed in traditional medicine for their anti-inflammatory properties and are currently being investigated in cancer therapy [24]. Acridines are quinoline analogues exist in many natural products that exhibit antiparasitic and antibacterial properties and are widely studied as anticancer agents against leukemia [25]. Lisboa et al. tried to address the high toxicity of acridines that hampers their clinical usefulness by conjugation with thiophene moieties (Figure 3) [26]. The toxicity and the biological properties of the new acridine–thiophene conjugates were subsequently investigated. Conjugate **6** demonstrated low micromolar IC_50_ value against HCT-116 cell lines and significantly reduced the tumor volume and cell viability of EAC xenografted mice compared to the control group. Acute preclinical toxicity assay demonstrated the survival of animal species after 300 mg/kg and then 2000 mg/kg of **6** treatment, while the estimated LD50 was higher than 5000 mg/kg. Therefore, the conjugation of acridine with thiophene moiety increased its antitumor activity and decreased toxicity.

Many conjugates containing thiazole, pyrrole and piperazine bioactive moieties are currently under clinical testing, due to the anticancer properties they exert. Voreloxin (**7**) is a quinolone–thiazole conjugate capable of inhibiting topoisomerase II and causing site-selective DNA damage (Figure 3). Voreloxin is being examined in two active clinical trials as a co-drug against acute myeloid leukemia (phase II) and myelodysplastic syndrome (phase I) [27]. Sunitinib (**8**) is an isatin (**10**) and pyrrole conjugate that inhibits multiple receptor tyrosine kinases and is already an FDA-approved drug against renal cell carcinoma, gastrointestinal stromal tumor, meningioma and pancreatic neuroendocrine tumor (Figure 3). Currently, sunitinib is being evaluated against other types of cancer, such as thymoma, glioblastoma and osteosarcoma, as a single drug or co-drug (phase II) [28]. Another isatin and piperazine conjugate is nintedanib (**9**), an idiopathic pulmonary fibrosis and lung cancer FDA-approved drug (Figure 3). Nintedanib is also a multitargeting bifunctional compound that mainly targets nRTKs and RTKs [29]. Its antitumor properties are evaluated in ongoing phase II clinical trials for advanced pancreatic cancer. Quinazolines are quinoline-derived alkaloids, present in many existing drugs like Lapatinib (**11**), a quinazoline and furan conjugate (Figure 3). Lapatinib has been FDA approved since 2007 for treating breast cancer, and its potential has been studied in more than 10 active clinical trials that include combinational therapy or dose adaptation studies. For further quinoline- [30], coumarin- [31,32] and indole-based [33,34] hybrids, the reader is referred to recent literature reviews.

### 2.2. Hybrids of Natural Products with Phenyl Moieties

Histone deacetylase inhibitors (HDACi) constitute a class of cytotoxic agents that have demonstrated substantial in vitro antitumor effects in different types of cancer cells, including colon cancer, breast cancer and liver cancer [35,36,37]. Histone deacetylases are considered as great targets in cancer therapy, although HDACi’s efficacy against solid tumors is low. Inhibition of HDACs results in activation of STAT3, which is overexpressed in cancer and regulates cancer formation, development and metastasis [38]. Ren et al. proposed an interesting approach of inhibiting both protein targets by conjugating vorinostat (SAHA), a known HDACi with pterostilbene (PTE), a STAT3 inhibitor (Figure 4) [39]. Their most prominent compound **12** exhibited great in vitro antitumor activity (IC_50_ = 0.78 μM on MDA-MB-231 cells) and successfully inhibited both STAT3 (KD = 33 nM) and HDACs (IC_50_ = 23.5 μM). The PTE–SAHA conjugate also exhibited great antitumor effects in 4T1 allograft mice. At 21 days post treatment (30 mg/kg), the novel compound substantially diminished the tumor volume and weight (64% against control) compared to both parent drugs. This work presents a great method of enhancing HDACI’s efficacy against solid tumors by including another pharmacophore capable of targeting a different pathway and inducing cell death.

Cancer cells overexpress heat shock protein 90 (Hsp90), reflecting their need to maintain protein homeostasis during stress conditions [40,41,42]. Hsp90s are drug targets in cancer therapy, but most Hsp90 inhibitors failed in clinical trials due to liver, ocular and cardiac toxicities [43,44]. Chae and his colleagues tried to address the existing interplay between Hsp90 and HDAC, as HDAC is believed to be vital for the development of resistance towards Hsp90 inhibitors, suggesting that the dual inhibition of Hsp90 and HDAC might overcome the issues that single inhibition of Hsp90 pose (Figure 4) [45,46]. Computational drug design was initialized based on already existing inhibitors, such as onalespib, ganetespib, tubastatin-a and their known pharmacophores, to synthesize a novel HDAC6/Hsp90 inhibiting hybrid **13**. This hybrid included a resorcinol moiety, found in many secondary metabolites, which is a very important pharmacophore. Its in vitro properties were tested in Hsp90α and HDAC6 assays and H1975 cells with IC_50_ values of 61 nM, 106 nM and 1.7 μM, respectively. Chae et al. also tested the hybrid’s antitumor properties in H1975 xenografted mice. Intraperitoneal administration of 25 and 50 mg/kg showed a delayed tumor growth, with a low inhibition rate and no significant side effects.

The recent literature demonstrates that the combination of immunotherapy and chemotherapy offers improved clinical benefits, deriving from targeting multiple cell survival pathways to overcome drug resistance [47,48,49]. Indoleamine-2,3-dioxygenase 1 (IDO1), which stimulates organism’s immune response, and signal transducer and activator of transcription 3 (STAT3), are targets to resolve MDR. Huang et al. synthesized a naphthoquinone conjugate **14** that exhibited mild nanomolar activity against STAT3 and IDO1 assays and antiproliferative effects against colon, ovarian, lung and hepatoma cancer cells with IC_50_ values in the range 0.012 μΜ to 0.033 μΜ (Figure 4) [50]. It also inhibited tumor growth in vivo in xenografted mice with a 53.4 and 59% inhibition rate (10 and 20 mg/kg per 3 days for 3 weeks, respectively), responding to both immunocompetent and nude mice. Thus, a series of new natural product-based bifunctional compounds were synthesized, exerting synergistic effects by targeting two different pathways and addressing drug resistance. Hybrids of natural products with phenyl moieties addressing tubulin inhibition [51] and multi-drug resistance [52] were also extensively reviewed in the recent literature.

### 2.3. Hybrids of Natural Products with Small Molecule Proteolysis Inducers

The conjugation of natural products with small molecules that can bind to ligases and induce proteolysis gave rise to the protein-targeted degradation strategy and the development of protein-targeting chimeras (PROTACs). PROTACs have a catalytic mode of action that allows for an efficient inactivation of a protein of interest and target undruggable proteins [53]. Recently, a PROTAC that recruits cereblon E3 ubiquitin ligase was synthesized by Hatcher et al. Initially, JH-VIII-49 was developed as a selective inhibitor of cyclin-dependent kinase 8 (CDK8) inspired by Cortistatin, A. JH-VIII-49 was linked to pomalidomide (a cereblon E3 ubiquitin ligase ligand) producing JH-XI-10-02 (**15**), which promoted proteasome degradation of CDK8 in Jurkat cells (Figure 5) [54]. In the same year, another natural product, wogonin, was utilized in PROTACs. Wogonin is derived from Scutellaria baicalensis and is a selective inhibitor of CDK9. Bian et al. synthesized a series of wogonin-based PROTACs targeting CDK9, using pomalidomide as a ligand of cereblon (Figure 5) [55]. Western blot analysis in MCF-7 cells indicated that one of the PROTACs, **16**, selectively degraded CDK9, contrary to wogonin, which inhibits all CDKs. Moreover, **16** demonstrated remarkable cytotoxicity exclusively against CDK9-overexpressing cancer cells.

Another natural product-based PROTAC was developed using pseudolaric acid B, which is isolated from golden larch bark and acts as an antagonist of CD147 (Figure 5). CD147 is a transmembrane glycoprotein with high expression levels in melanoma cells. In 2020, Zhou et al. conjugated PAB with derivatives of thalidomide to generate PROTACs targeting CD147 [56]. More specifically, **17** was observed to suppress the proliferation of Sk-Mel-28 cells (IC_50_ = 6.23 μM) and induced CD147 degradation (DC_50_ = 6.72 μΜ) through a ubiquitin–proteasome mechanism. In addition, the in vivo experiments in BALB/c female nude mice revealed not only the reduction of CD147 levels, but also a successful decrease in volume and weight of tumors.

Ursolic acid (UA), which is a pentacyclic anticancer triterpenoid, has also been used for the preparation of PROTACs. Six PROTACs were synthesized by Qi et al. by conjugating UA with thalidomide (ligand for E3 ligase cereblon) via different linkers, and their mechanism of action was investigated [57]. The results showed that compound **18** had noteworthy antitumor activity in vitro (IC_50_ = 0.23 μM against A549, Huh7, HepG2) and decreased the levels of MDM2 protein (Figure 5). MDM2 appeared to be a target of UA, and the antiproliferative effect of the synthesized PROTAC is a consequence of the MDM2/P53/P21 pathway. Further future research is required to clarify if UA PROTACs could exhibit dual degradation activity, since MDM2 is an E3 ubiquitin-protein ligase.

Pomalidomide was employed in the synthesis of a novel conjugate of an indirubin analogue (an indole alkaloid from Indigofera species), which is a selective HDAC6 inhibitor by Cao et al. (Figure 6) [58,59]. HDAC6 has been related with many abnormalities including tumorigenesis due to its multiple roles in cells. PROTAC **19** induced selective degradation of HDAC6 (DC_50_ = 108.9 nM), promoting an upregulation in α-tubulin acetylation in K562 cells in a time-dependent manner. In addition, PROTAC **19** reduced NLRP3 inflammasome activation in LPS-induced mice. Thus, treatment with **19** in an LPS-induced endotoxic shock mouse model reduced the levels of interleukin 1 beta (1L-1β) as a result of HDAC6 degradation.

Platanic acid, a nortriterpenoid isolated from the leaves of Syzigium clavrfium [60], stood out in a SPR study as an inhibitor of dysregulated yes-associated protein (YAP), which is related with various types of cancer. Several chimeras were synthesized from Nakano et al. in 2022, using platanic acid and ligand LCL-161, which binds to E3 ubiquitin ligase Ciap [61]. Among them, **20** displayed the most promising results (Figure 6). Although **20** showed weak degradation of YAP, the expression of CTGF mRNA (a direct target gene of YAP/TEAD signaling) decreased in a dose-dependent manner after treatment with **20** in a luciferase reporter assay. PROTAC **20** demonstrated cytotoxic activity against NCI-H290 cells, where YAP is overexpressed, while it did not affect cells where YAP is expressed in low concentrations like MeT-5A and A549 cells.

The conjugates presented above used natural products exclusively as ligands of the protein of interest (POI) in targeted protein degradation (TDP). However, most recently, a few examples reveal a different role of some natural products that of ligase recruiters. Nimbolide is a limonoid terpenoid, originated from the neem tree. It exerts its antitumor activity through an 1,4-unsaturated group that interacts with C8 of RNF114, which has an E3 ligase function. In 2019, Spradlin et al. synthesized two PROTACs using nimbolide as an E3 ligase ligand, a linker and the BRD4 inhibitor JQ1 [62]. One of them, PROTAC **21**, turned out to induce dose- and proteasome-dependent degradation of BRD4 selectively, at a concentration of 0.1 μM after 12 h against 231MFP cells (Figure 7). In another study, the same research group attempted to target BCR-ABL by linking nimbolide with dasatinib (a BCR-ABL inhibitor) (**22**) (Figure 7) [63]. The results indicated a selective degradation of c-ABL from **22**, in contrast to previous cereblon- or VHL-recruiting degraders. Τhese developments rendered the RNF114 protein as a novel ligase candidate that could be crucial in TPD [64]. In addition, Bardoxolone is a derivative from oleanolic acid triterpene that interacts reversibly with cysteines in KEAP1. Kelch-like ECH-associated protein 1 (KEAP1) is a component of the Cullin 3 E3 ubiquitin ligase complex, which regulates the action of NRF2 [65]. Therefore, bardoxolone could act as a covalent ligase recruiter. In 2020, Tong et al. reported a hybrid of bardoxolone with JQ1 (**23**) (Figure 7). The PROTAC promoted dose-, proteasome- and NEDDylation-dependent degradation of BRD4 in the 231MFP cell line (100–200 nM) [66].

Piperlongumine (PL) is a natural product originated from the plant Piper longum L. possessing multiple applications in drug discovery [67,68]. An interesting function of PL has been reported recently as a ligase recruiter. More specifically, in 2022, Pei et al. synthesized a series of PROTACs with PL and a CDK9 inhibitor, SNS-032. One of them (**24**) demonstrated proteasome-dependent degradation of CDK9 (DC_50_ = 9 nM after 16 h) in three cancer cell lines (MOLT4, 293T, K562) (Figure 7) [69]. Additionally, KEAP1 appeared to be the target E3 ligase of PL by using ABPP and TurboID-bait assays. To evaluate the activities of the new ligand, a hybrid of PL with Ceritinib was synthesized, targeting EML4-ALK fusion protein (Figure 7). The new bivalent molecule **25** promoted degradation of EML4-ALK in a concentration-dependent manner. These results expand the boundaries in TPD, since the small molecular weight of PL may help to overcome existing problems in pharmacokinetics of PROTACs.

### 2.4. Hybrids of Natural Products with Small Molecules Autophagy Inducers

Natural products have also been coupled to small molecules’ autophagy inducers to create autophagy-targeting chimeras (AUTOTACs) for cancer therapy. During autophagy, cells recycle cytoplasmic ingredients including proteins, aggregates and organelles into autophagosome for lysosomal hydrolysis [70]. Protein p62, also known as sequestosome-1(SQSTM1), serves as an autophagy receptor and delivers ubiquitinated proteins to the proteasome for degradation [71]. The natural product fumagillin was employed for the preparation of AUTOTACs from Ji et al. in 2022 [72]. It was conjugated through a PEG linker with YTK-105, a ligand of p62, that plays a crucial role in the formation of autophagosome as previously mentioned (Figure 8). The newly synthesized AUTOTAC **26** appeared to promote p62 self-polymerization and degradation of MetAP-2 (DC~0.7 μΜ in HEK293 cells and DC~500 nM in U87-MG glioblastoma cells) via autophagosomes. In addition, AUTOTAC **26** could reduce cancer cell proliferation and induce apoptosis, offering a new perspective in TPD.

### 2.5. Folate Hybrids of Natural Products

Targeted delivery strategies increase a drug’s concentrations in tumor sites leaving the healthy cells unaffected, thus leading to high efficacy and reduced side effects [73]. Folate hybrids of natural products have been developed in the field and have shown high targeting ability. Folic acid binds to folate receptors, which are cell-surface glycoproteins, belonging to that FR family, which consists of four members FRα, FRβ, FRγ and FRδ [74,75]. FRs are overexpressed on many epithelial cancers and tumors such as carcinoma, breast and brain tumors.

Scaranti et al. reported some folate-targeting conjugates consisting of folic acid and natural cytotoxic agents [76]. Such conjugates are the EC131 (**27**), the EC145 (**28**) and the BMS-753493 (**29**) (Figure 9). EC131 consists of the maytansinoid DM1 coupled with folic acid through a disulfide bond and exhibited marked antitumor activity in s.c. FR-positive M109 tumors in BALB/c mice with no evidence of systemic toxicity. EC145, known as vintafolide, is a folic acid analogue with high water solubility containing a peptide spacer, linking it to the desacetylvinblastine monohydrazide (microtubule-destabilizing agent). It was successfully tested in Phase I and II clinical trials for the treatment of ovarian cancer with encouraging results and an acceptable safety profile that led to a Phase III trial, which was discontinued in 2014. Later, it was evaluated in patients with small cell lung cancer in a Phase II trial that highlighted the enhanced antitumor efficacy of the vintafolide and docetaxel combination. BMS-753493 (**29**) is a different folate hybrid of epothilone A that underwent two parallel Phase I/IIa studies, which were also discontinued due to the lack of objective responses. Finally, EC1456 (**30**), a folate–tubulysin conjugate, was developed in 2018 by Reddy et al. (Figure 9). It demonstrated substantial specificity and antiproliferative activity in preclinical models of FR-positive tumors; therefore, it was advanced in a Phase I trial [77].

### 2.6. Steroidal Hybrids of Natural Products

Conjugation of steroids with other bioactive compounds, especially anticancer agents to develop bifunctional molecules, is extensively studied. The resulting conjugates possess altered physicochemical properties, higher selectivity towards cancer cells and improved cytotoxic activity [78,79]. Steroid receptors are overexpressed on the membrane of cancer cells, especially in hormone-dependent tumors, thus posing an ideal target for drug delivery [80]. Moreover, studies have shown that the steroidal moieties in conjugates provide a synergistic effect, enhancing the antitumor activity of the delivered drug [81]. In 2019, Ke et al. reported a synthesis of a series of steroidal hybrids of isatin (**31**), employing epiandrosterone and androsterone and evaluated their anticancer activity in three different cancer cell lines, compared to the chemotherapeutic 5-fluorouracil (5-FU) (Figure 10) [82]. The results showed the hybrids’ superior cytotoxic activity compared to 5-FU, with low to moderate IC_50_ values (2.39–42.21 μΜ). 17α-substituted testosterone and 17β-substituted epitestosterone conjugates with pyropheophorbide (**32**) through different linkers were synthesized by Zolotsev and his group, and their efficacy was investigated against two prostate cancer cell lines (Figure 10) [83]. The epitestosterone conjugates were internalized more efficiently than the testosterone ones, a feature that was attributed to the structural difference between the steroidal moieties. Furthermore, MTT data revealed a strong antiproliferative effect against both cell lines after 96 h of incubation, although the effect in LNCaP cells was more pronounced.

Chaikomon et al. reported the conjugation of doxorubicin to dexamethasone (**33**) and the apoptotic effect of this hybrid to MCF7 cells (Figure 10) [84]. The hybrid showed an altered mode of action compared to doxorubicin and better cellular penetration of doxorubicin, which is possibly due to the lipophilic properties of dexamethasone. The novel compound induced cell death through ROS generation, but its antiproliferative effect compared to doxorubicin alone was lower. However, it is noteworthy that it could overcome P-gp efflux in MDR-1-overexpressing cells and exert its cytotoxic activity.

### 2.7. Hybrids Incorporating Two Natural Products

Natural products have been conjugated with other secondary metabolites to form bifunctional compounds. Polyphenols like curcumin (Cur) and resveratrol (Res) have been extensively studied in conjugation with other polyphenols for their anticancer activity. Like most polyphenols, Cur and Res have poor water solubility and quick metabolization in slightly alkaline pH, resulting in non-encouraging results in preclinical models and clinical trials [85,86]. The importance of conjugation is highlighted in a recent review of Micale, summarizing various Cur and Res conjugates that interact with more than one cellular target and exhibit increased efficacy compared to parent drugs [87].

Magnolol (Mag) is a lignan isolated from *Magnolia officinalis* that has anti-inflammatory, antioxidant and anticancer properties [88]. Sulforaphane (Sfn) is a very potent cytotoxic agent that belongs to the isothiocyanate secondary metabolite family [89]. Tao and his colleagues were the first to synthesize Mag–Sfn hybrids to study their biological properties (Figure 11) [90]. The novel compounds CT1-2 (**35**) and CT1-3 (**36**) were prepared, and their anticancer activity was evaluated against H460, LOVO and MCF-7 cells. The results revealed that CT1-3 was the most potent inhibitor acting in the mid micromolar range. CT1-3 was further tested in animal models. where it effectively reduced tumor growth. More specifically, MDA-MB-231 xenografted mice were treated with CT1-3 (20 mg/kg), and after 28 days, it significantly inhibited tumor growth (above 66%), compared to the control group. Finally, it was shown that CT1-3 displayed its action via mitochondria-mediated apoptosis and inhibition of epithelial mesenchymal transition.

A recent work presented by Ma et al. demonstrated the synthesis and biological evaluation of ARTD, a novel artemisinin and daumone hybrid (**37**) (Figure 12). Artemisinin is a sesquiterpene lactone that displays cytotoxic effects against drug-sensitive and drug-resistant cell lines [91,92,93,94]. Daumone, on the other hand, is a biodegradable glycolipid with low toxicity that can improve the solubility, as well as the activity of the hybrid system [95]. The primary focus of the study was the inhibition of bone metastasis of cancer cells, which was evaluated by viability, migration, invasion, transwell invasion and wound-healing assays. Compared to artemisinin alone, ARTD successfully inhibited cancer cell-mediated osteolysis as it blocked the induction of severe osteolytic lesions in mice tibia bone. This was achieved by targeting the tumor-suppressive protein ATF3 and oncogenic protein E2F1 [96].

Formononetin is an *O*-methylated isoflavone isolated from the red clover. By regulating the mitogen-activated protein kinase signaling pathway, formononetin exerts a variety of anticancer, anti-inflammatory and antioxidant effects [97,98,99]. Yao et al. designed a new hybrid (**38**) by combining formononetin with umbelliferone, a potent 7-hydroxy coumarin (Figure 12) [100]. Alkynyl and azide chains were attached to formononetin and umbelliferone, respectively, by common etherification reactions, and the final hybrid was produced after 1,2,3-triazole moiety bridging. The novel hybrid was tested against three gastric cancer cell lines where a low micromolar IC_50_ was achieved against SGC7901 cell line (compared to mid micromolar activities of its parent compounds). In addition, it inhibited migration by interacting with the Wnt/β-Catenin and Akt/mTOR pathways. The hybrid effectively inhibited SGC7901 xenograft tumor growth in vivo without significant toxicities to other organs. Specifically, 100 mg/kg doses were used both for hybrid and control, with the former demonstrating a 71.7% inhibitory rate after 21 days.

A series of β-carboline dimers linked by *N*-acylhydrazone were synthesized by Guo et al. β-carbolines are natural alkaloids possessing a broad spectrum of biochemical activities including antimicrobial and antitumor properties [101,102,103]. Two substituted β-carboline subunits were linked together by an acylhydrazone moiety to form a novel hybrid (Figure 12). 2,3,4,5,6-perfluorophenylmethyl substituted hybrid **39** was the most potent derivative that inhibit cancer cell proliferation in various cancer cells. It was also evaluated against sarcoma 180 xenografted mice displaying a 40% tumor growth inhibition and possessing significant anti-angiogenic and antimetastatic properties [104]. Hybrids incorporating two natural products and their anticancer properties were also extensively reviewed by Choudhary et al. [105].

## 3. Antibody–Drug Conjugates Based on Natural Products

Monoclonal antibodies are well established in clinical use as therapeutic agents since they exhibit high specificity and affinity to multiple biochemical targets. They are employed as targeting agents in drug delivery systems to accumulate a drug to specific diseased cells avoiding healthy tissues [106]. Antibody–drug conjugates (ADCs) are a rapidly growing field of pharmaceuticals [107]. Currently, over 80 ADCs are in clinical development [108], while 9 have been approved by the FDA and 4 of them are approved by the EMA [109]. In 2020, Hafeez et al. reviewed all the FDA-approved ADCs and the ones that are in ongoing clinical trials [110]. The cytotoxic agents used in the FDA-approved conjugates are calicheamicin derivatives, mertansine (DM1), monomethyl auristatin E and F (MMAE and MMAF, respectively) and two camptothecin analogues (SN-38 and DX-8951). The antibodies conjugated to them are all humanized to prevent mounting an immune response against them. Most cytotoxic agents found in ADCs in ongoing clinical trials belong to the maytansinoid and dolastatin families that have shown significant anticancer activities in preclinical studies.

Trastuzumab emtansine (**40**) is a conjugate that consists of the monoclonal antibody anti-HER2 and the tubulin inhibitor DM1 (mertansine) (Figure 13). It was approved in 2013 by the FDA after the impressive results of two Phase III trials in patients with ErbB2-positive advanced breast cancer [111,112]. In both studies, patients had extended survival times compared to the conventional therapies administered with anticipated adverse effects. At present, trastuzumab emtansine is in 36 ongoing Phase I–III trials in ErbB2-positive breast, lung and colorectal cancer as well as other solid tumors. Brentuximab vedotin (**41**) received FDA approval in 2011 for relapsed or refractory systemic anaplastic large-cell lymphoma (Figure 13) [113]. It is an anti-CD30-MMAE conjugate currently evaluated in 83 Phase I–III clinical trials in various hematological malignancies in different treatment combinations. Inotuzumab ozogamicin (**42**) is an anti-CD22 monoclonal antibody linked to a calicheamicin derivative (*N*-acetyl gamma-calicheamicin-dimethyl hydrazide), which received FDA approval in 2017 for treatment of relapsed or refractory CD22+ acute lymphoblastic leukemia after a Phase III clinical trial, although severe side effects were observed (Figure 13) [114]. Results from 19 ongoing clinical trials in lymphoblastic leukemias are anticipated.

Gemtuzumab ozogamicin (**43**) is an antibody–drug conjugate similar to Inotuzumab ozogamicin, consisting of the anti-CD33 instead of anti-CD22 antibody. ADC **43** carries the same calicheamicin analogue and linker as ADC **42** and gained accelerated FDA approval in 2000 for the treatment of CD 33+ acute myeloid leukemia (AML). Compound **43** was soon withdrawn due to severe toxicities, and it was reapproved in 2017 after reliable data that ensured safety for patients [115]. Its antitumor efficacy in patients is currently investigated in 31 active trials for AML, acute promyelocytic leukemia and myelodysplastic syndromes.

Polatuzumab vedotin (**44**) is an ADC consisting of an anti-CD79b monoclonal antibody coupled to the same cytotoxic agent MMAE and linker like Brentuximab vedotin **41**. It was approved in 2019 by the FDA for treatment of relapsed or refractory diffuse large B cell lymphoma (DLBCL). Its main adverse effects are myelosuppression, peripheral neuropathy and progressive multifocal leukoencephalopathy [116]. A recently reported Phase III trial evaluated a modified regimen, which employed polatuzumab vedotin instead of vincristine, in patients with previously untreated intermediate-risk or high-risk DLBCL. The differences between the standard regimen and the regimen that included Polatuzumab vedotin were not statistically significant [117]. Polatuzumab vedotin is also investigated in 8 Phase I–III trials in hematological malignancies. Trastuzumab deruxtecan or Enhertu (**45**) is another ADC based on the anti-HER2 monoclonal antibody conjugated with exatecan, a camptothecin derivative, with a different drug to antibody ratio compared to trastuzumab emtansine (Figure 14). It is administered to previously treated ErbB2-positive metastatic breast cancer patients and received FDA approval in 2019. Enhertu is associated with only a few side effects such as myelosuppression, nausea and interstitial lung disease [118].

MMAE has been also utilized in the manufacture of Enfortumab vedotin (**46**) (Padcev) a Brentuximab vedotin **41** analogue. In the case of ADC **46**, MMAE is conjugated with a nectin-4-directed monoclonal antibody via the same linker. Padcev was approved in 2019 by FDA for treatment of patients with relapsed or refractory locally advanced or metastatic urothelial cancer, after a Phase II clinical trial with impressive results. However, some serious adverse effects were observed with febrile neutropenia being the most common of them [119]. Padcev is currently in seven Phase I–III trials, and its efficacy has been investigated compared to chemotherapy and in combination with immunotherapy in patients with advanced urothelial cancer.

Sacituzumab govitecan (**47**) is a trophoblast cell surface antigen 2 (TROP2)-targeting ADC coupled with SN-38, a camptothecin analogue, that has been approved in 2019 for patients with metastatic triple-negative breast cancer (Figure 14) [120]. In a single-arm Phase I/II trial, it showed significantly prolonged survival time with very mild side effects compared to other ADCs. In present, it is investigated for its therapeutic effect in clinical trials for other types of cancer like endometrial cancer, glioblastoma and urothelial cancer.

Finally, Belantamab mafodotin (**48**) is the most recently approved ADC, comprising the humanized IgG1κ monoclonal antibody conjugated with a member of the auristatin family, maleimidocaproyl monomethyl auristatin F (MMAF) (Figure 14). This ADC targets BCMA, a cell surface B cell maturation antigen expressed on multiple myeloma cells, which promotes growth and survival of plasma cells [121]. The Phase II study, which gained the approval of belantamab mafodotin in heavily pretreated patients with multiple myeloma, exhibited response rates ranging from 31 to 34% after dosages of 2.5 mg/kg and 3.4 mg/kg every three weeks [122]. Currently, it is investigated in five other ongoing Phase I–III clinical trials.

Another ADC that has been studied in clinical trials is disitamab vedotin (RC-48) (**49**) (Figure 15). This conjugate consists of HER2 antibodies coupled via a valine-citruline linker with the maytansinoid MMAE and targets different epitopes of the HER2 receptor. The HER2 antibodies bind to HER2 receptors on cancer cells with high affinity, and the conjugate enters the cell through clathrin- and caveolin-mediated endocytosis. Upon endocytosis, the cytotoxic tubulin depolymerization agent MMAE is cleaved by activated lysosomal enzymes and induces mitotic cell cycle arrest and apoptosis [123]. The novel ADC has been tested in Phase I and II human trials for breast carcinoma, urothelial cancer and gastric carcinoma/gastroesophageal junction adenocarcinomas with evident antitumor activity and good survival rates. Regarding the safety profile, approximately 94.7% of patients experienced common side effects after the first 2 days of treatment, while the most common grade 3 or worse side effects were observed only in the high-dose groups [124,125,126]. Currently, 12 other clinical trials of Phase I–III are in progress testing RC-48′s efficacy in a variety of cancers such as breast, gastric, urothelial, biliary tract and non-small cell lung cancer [127].

Poudel et al. reported the synthesis of antibody–drug conjugates for the delivery of uncialamycin and tested their in vivo activity in H226 human lung carcinoma xenograft model in mice (Figure 15) [128]. The antibodies chosen in this study were anti-CD70 and anti-mesothelin, which bind to overexpressed receptors in renal and lung cancer cells, respectively. The most potent conjugate was the one with anti-mesothelin, which exhibited high targeting ability and complete tumor growth inhibition for 6 weeks after a single dose.

## 4. Peptide Hybrids with Natural Products

Peptides are one of the most prominent types of carriers used in molecular drug delivery systems as they are characterized by precise tumor-targeted delivery and release of the drug they are conjugated to. Peptide–drug conjugates (PDCs) selectively bind to membrane receptors of tumor cells and are uptaken by the cells through transmembrane effects such as endocytosis and internalization [129]. Such hybrids have successfully been examined in recent clinical trials with promising results. Mipsagargin (**51**) is a prodrug consisting of a thapsigargin analogue conjugated via an amide bond with the PSMA substrate-masking peptide Asp-γ-Glu-γ-Glu-γ-Glu-Glu (Figure 16) [130]. PSMA is a cell surface transmembrane glycoprotein highly expressed by hepatocellular tumors, which can cleave the Glu residues in the masking peptide to deliver thapsigargin specifically to the tumor site. This prodrug was extensively studied in a Phase II clinical trial in 2019 for patients with advanced refractory hepatocellular carcinoma, following a preliminary Phase I study in 2016 [131]. The results of this trial indicated that the highest tolerated dose of the prodrug (40 mg/m^2^) led to disease stabilization and decreased tumor blood flow in metastatic sites with a relatively good safety profile, warranting a future larger clinical study for mipsagargin’s activity [132]. In another study, Whalen et al. reported the synthesis and biological evaluation of a miniaturized peptide–mertansine conjugate (PEN-221) (**52**) for the treatment of small-cell lung cancer (Figure 16). The conjugate was designed to target somatostatin receptor 2 (SSTR2) through the peptide, which is a SSTR2 agonist [Tyr3, Cys8] octreotate. The results showed that the peptide indeed binds strongly and selectively to SSTR2 receptors, and its antiproliferative activity is directly dependent upon binding. In addition, PEN-221 induced apoptosis as evidenced by the increased levels of cleaved caspase-3. The in vivo experiments revealed a significant antitumor activity of PEN-221 with a maximum tolerated dose of 2mg/kg once a week that resulted in complete regressions in SSTR2-positine small-cell lung cancer xenograft mouse models [133].

In 2020, Bennett’s group reported the synthesis of a novel conjugate consisting of a bicyclic peptide linked via a cleavable linker to the auristatin derivative MMAE (**53**) (Figure 17) [134]. This peptide targets the erythropoietin-producing hepatocellular receptor EphA2, one of the members of the Eph family of receptor tyrosine kinases [135]. In vivo experiments in mice, rats and monkeys revealed a high selectivity and binding affinity to HT-1080 cells with a long drug-release rate, low uptake by normal cells and rapid excretion through kidney to bladder. Moreover, the new hybrid not only suppressed growth in solid tumors, but also in small metastatic tumors in mice implanted with PC3 xenografts, with extended survival times compared to the control group. Another bicyclic peptide was utilized in the study of Gowland et al. for the targeted delivery of mertansine to membrane type I matrix metalloproteinase (MT1-MMP) in an ongoing Phase I/IIa human clinical trial for the treatment of advanced solid tumors (Figure 17). The first phase of this trial demonstrated the superior pharmacokinetic profile of the hybrid (**54**), while more data are expected upon completion of the clinical trial [136].

Until today, the most progressed peptide–drug hybrid system for selective chemotherapy was the FDA-approved Zoptrex (**55**), a GnRH analogue coupled with doxorubicin via an ester bond with a glutaric acid spacer (Figure 18). Although it reached a Phase III clinical study for the treatment of endometrial cancer, the disclosed results of this study were not as encouraging as anticipated, as it did not extend the survival time of patients significantly more than doxorubicin alone [137]. On the other hand, another prominent hybrid for selective targeting is ANG1005, a BBB-penetrating angiopep-2 peptide conjugated with three paclitaxel molecules via ester bonds (**56**) (Figure 18). ANG1005 has successfully passed a Phase I clinical trial with very high tolerance, while Phase II and III results from clinical trials are yet to be published [138].

## 5. Aptamer–Drug Conjugate Based on Natural Products

Nucleic acid aptamers, often called “chemical antibodies”, are small single-stranded oligonucleotides characterized by significant selectivity and binding affinity towards their targets, including proteins and alive cancer cells. Owing to their advantages, like ease of synthesis and functionalization, high shelf-stability, low immunogenicity and ease of antidote development, aptamers have been extensively employed as carriers in targeted drug delivery [139]. In their work, Gray et al. demonstrated the synthesis of novel aptamer conjugates (ApDC) (**57**, **58**) that consist of E3, an RNA aptamer, and two auristatins, MMAE and MMAF, and evaluated their antitumor activity against prostate cancer (Figure 19) [140]. Both conjugates induced cell death in all prostate cell lines with IC_50_ values ranging from 2.3 to 152 nM. Treatment of mice with MMAE-E3 (1.03mg/kg) exhibited strong growth inhibitory effect and increased survival rate, contrary to the non-active MMAF-E3 conjugate.

Recently, Jeong et al. developed a novel ApDC (**59**) for targeted therapy in breast cancer (Figure 19). This agent consists of a HER2 RNA aptamer conjugated to mertansine [141]. This ApDC’s in vitro as well as in vivo antitumor efficacy was evaluated in HER2-positive BT-474 and HER2-negative MDA-MB-231 cancer cells and in mice carrying BT-474 breast tumors overexpressing HER2. Results indicated a specific and efficient binding affinity to HER2-positive BT-474 cells, but not to HER2-negative MDA-MB-231 cells. The same pattern of action was also observed in cytotoxic assays where the ApDC was not effective against the HER2-negative MDA-MB-231 cells, while it was moderately effective in HER2-positive BT-474 cells. The in vivo experiments revealed a superior inhibitory activity in tumor growth compared to mertansine after a dose of 60 μg/kg, with no evident systemic toxicity.

In 2020, He et al. reported the synthesis of a novel ApDC (**60**) comprised of the nucleolin-specific aptamer AS1411 coupled to the diterpenoid triptolide and demonstrated its impressive results in preclinical studies against triple-negative breast cancer (Figure 19) [142]. This new agent exhibited excellent specificity and superior in vitro and in vivo cytotoxic profile with a 67% reduction in tumor growth. It enhanced triptolide’s antitumor activity and caused negligible adverse effects, rendering it a perfect candidate for further clinical studies. Earlier studies of AS1411 conjugates with other natural cytotoxic agents, such as doxorubicin and paclitaxel, with noteworthy anticancer activity are included in a recent literature review [143].

## 6. Hybrids of Natural Products with Metal Complexes

The incorporation of an organometallic motif into drugs constitutes a strategy that has been utilized in the synthesis of anticancer drugs since the beginning of the 20th century. Ferrocifen, a ferrocenyl analogue of tamoxifen, and cisplatin are characteristic examples of this strategy. Unique characteristics of metals, such as redox activity, variable coordination modes and reactivity with several organic substrates, can improve the pharmacokinetic profile and the antitumor efficacy of novel bioactive compounds when incorporated into their scaffold [144]. Lapachol is a naphthoquinone originated from *Bignoniaceae* plants with multiple biological applications. Oliveira et al. synthesized a complex (**61**) of lapachol with Ruthenium (II)/triphenylphosphine to enhance lapachols’ anticancer activity (Figure 20) [145]. The results suggested that complex cis-[Ru(PPh_3_)_2_(lap)_2_] had better cytotoxic activity against A549 and MDA-MB-231 cancer cells than cisplatin, while the normal cells were not affected, indicating cell selectivity.

Podophyllotoxin (PPT) is a pentacyclic lignan isolated from the roots of *Podophyllum peltatum* with well-documented antitumor activity. Beauperin et al. conjugated the antimitotic PPT with a ferrocenyl moiety (**62**) to eliminate the side effects of PPT and convert it into a selective anticancer agent (Figure 20) [146]. Although the complex appeared to be less cytotoxic against MDA-MB-231 and MCF-7 cancer cells (IC_50_ 0.43 and 0.93 μM) than PPT (IC_50_ = 0.01 μM), the addition of ferrocenyl moieties is proposed as a method to achieve selective targeting of tumor cells, since its reversible redox behavior could interact with the cell’s microenvironment.

## 7. Conclusions

Natural products and their derivatives are structurally complex chemical compounds, consisting of aromatic, heterocyclic or aliphatic parts possessing extraordinary bioactivities, such as analgesic, anti-inflammatory, antibiotic, antifungal and anticancer properties [147]. Since ancient times, they play a crucial role in drug discovery for the development of pharmaceutical products for multiple diseases, like cancer [148]. Drawbacks of their activity in terms of physicochemical characteristics can be resolved by numerous strategies, including conjugation with other bioactive moieties (Table 1). The novel hybrids appear to be more potent, selective and able to target more than one biochemical pathway. In fact, most of the hybrids mentioned in this work have shown remarkable results in clinical trials, while some of them have already received FDA approval. In future, there is much to expect in the emerging fields of biopharmaceuticals and PROTACs. Biopharmaceuticals based on antibodies may avoid many of the side effects of classic chemotherapy and facilitate personalized patient treatments. As the technology advances further, PROTACs give a new perspective in drug discovery, allowing for a more efficient targeting of so far ‘undruggable’ proteins, yet their efficacy in clinic remains to be validated.

## Figures and Tables

**Figure 1 molecules-27-06632-f001:**
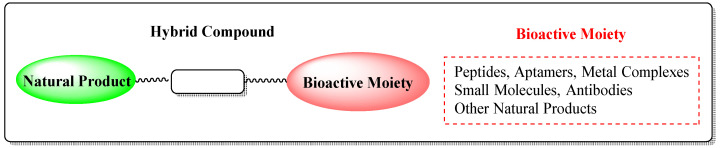
General structure of a natural product-based hybrid.

**Figure 2 molecules-27-06632-f002:**
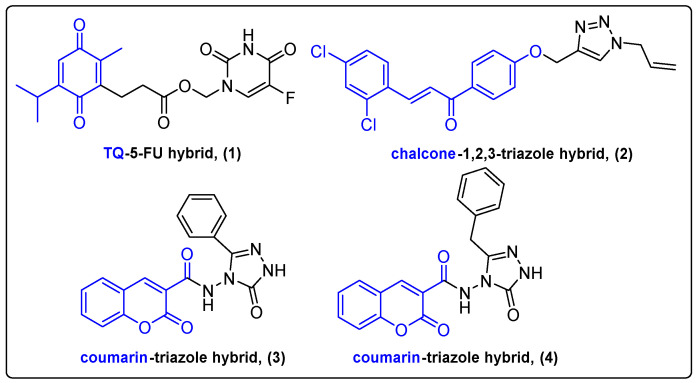
TQ, chalcone and coumarin hybrids.

**Figure 3 molecules-27-06632-f003:**
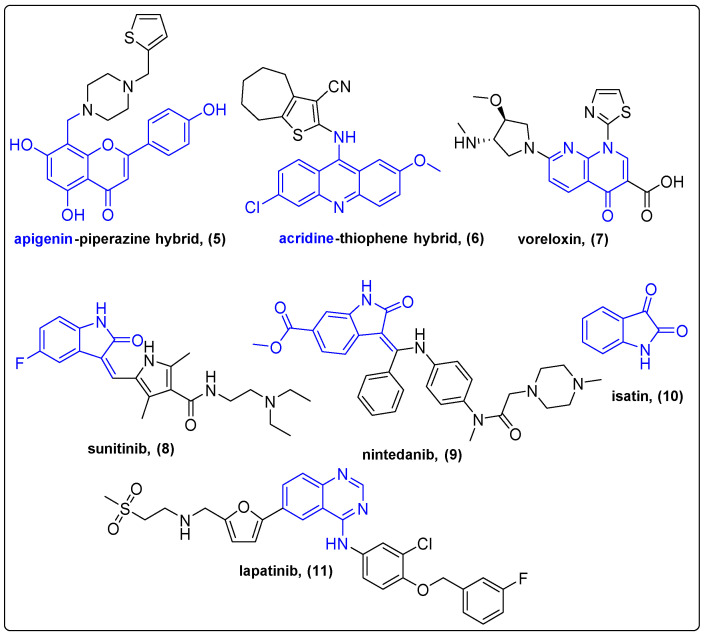
Natural products and their hybrids with heterocyclic moieties.

**Figure 4 molecules-27-06632-f004:**
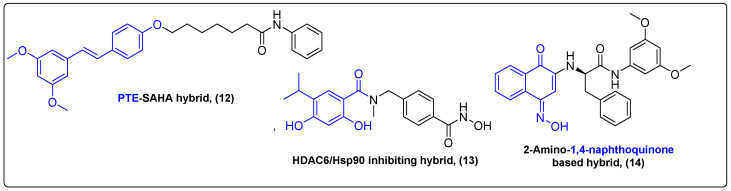
Natural product hybrids bearing phenyl moieties.

**Figure 5 molecules-27-06632-f005:**
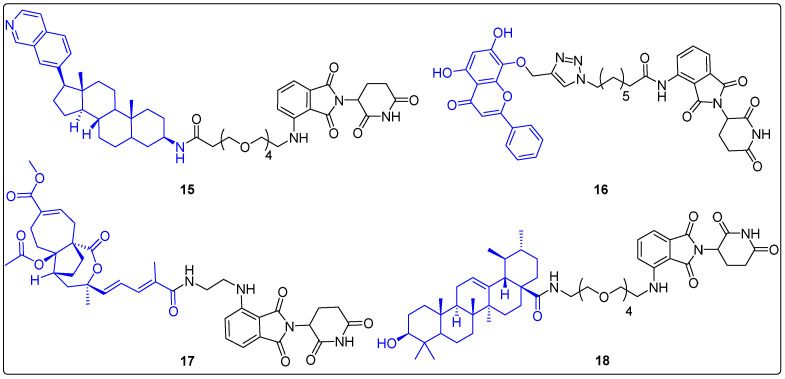
Natural products in PROTACs.

**Figure 6 molecules-27-06632-f006:**
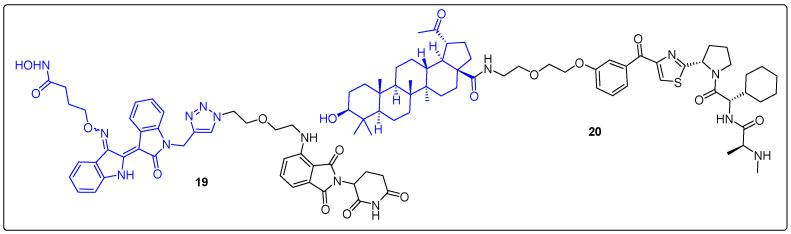
Natural product-based PROTACs.

**Figure 7 molecules-27-06632-f007:**
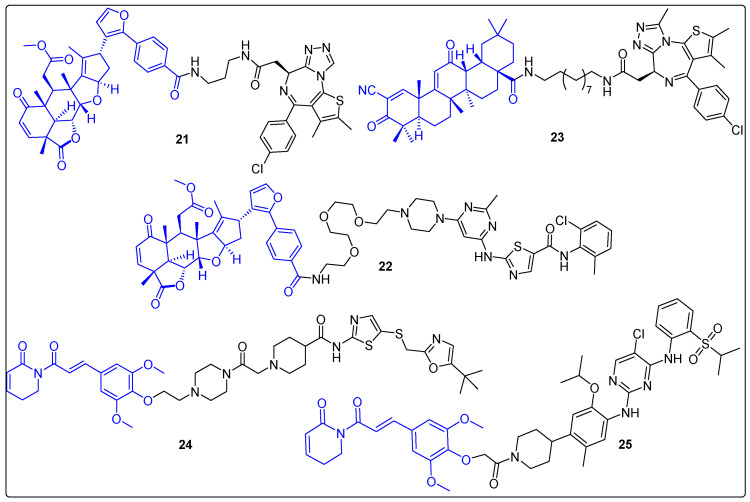
Natural products as E3 ligase recruiters and their PROTACs.

**Figure 8 molecules-27-06632-f008:**
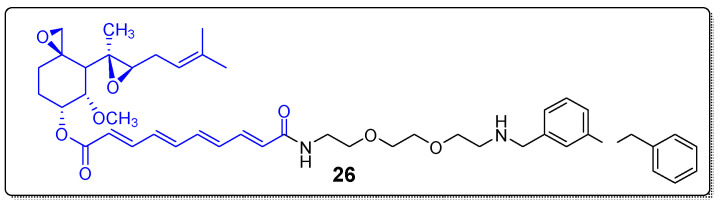
Fumagillin conjugated to TK-105.

**Figure 9 molecules-27-06632-f009:**
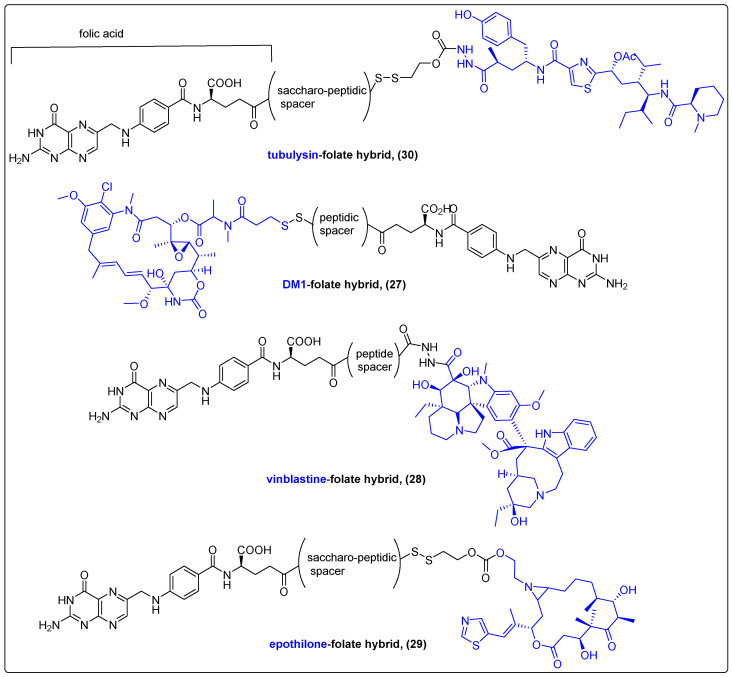
Folate hybrids of natural products.

**Figure 10 molecules-27-06632-f010:**
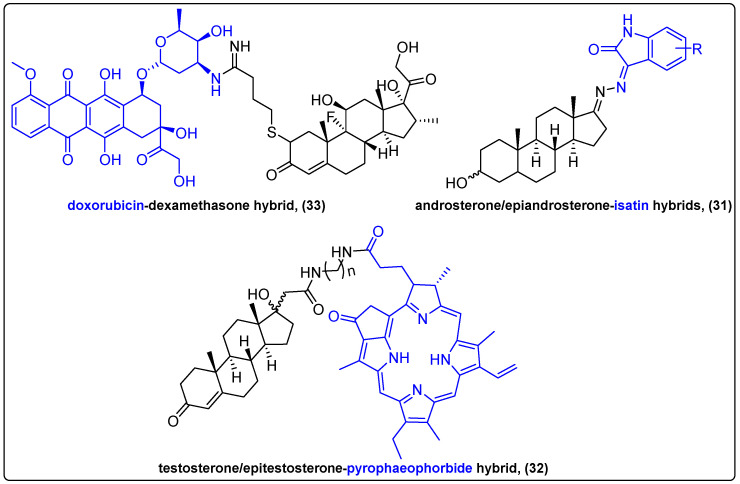
Steroidal hybrids of natural products.

**Figure 11 molecules-27-06632-f011:**
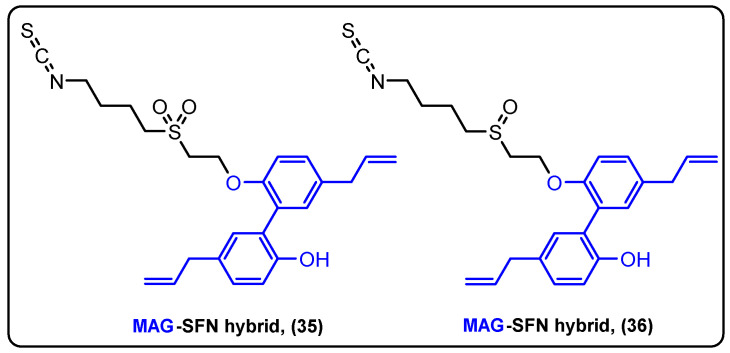
Magnolol–sulforaphane hybrids.

**Figure 12 molecules-27-06632-f012:**
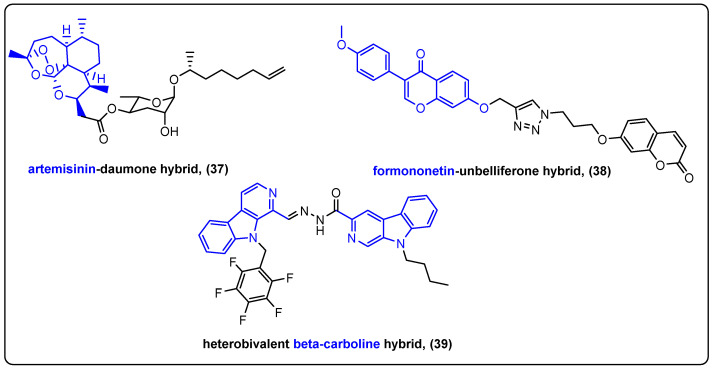
Hybrids incorporating two natural products.

**Figure 13 molecules-27-06632-f013:**
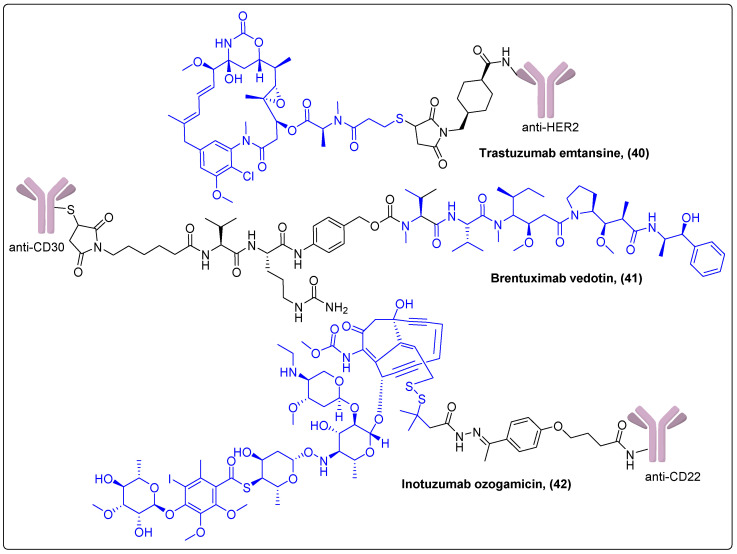
Antibody–drug conjugates carrying mertansine, MMAE or a calicheamicin derivative.

**Figure 14 molecules-27-06632-f014:**
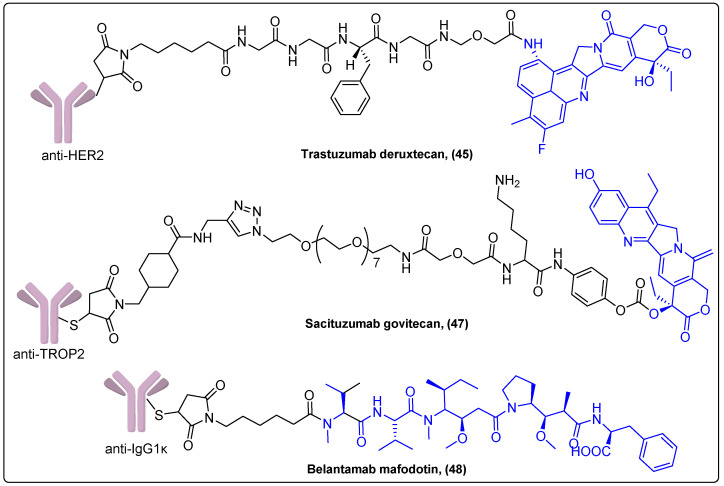
Antibody–drug conjugates carrying exatecan, SN-38 or MMAF.

**Figure 15 molecules-27-06632-f015:**
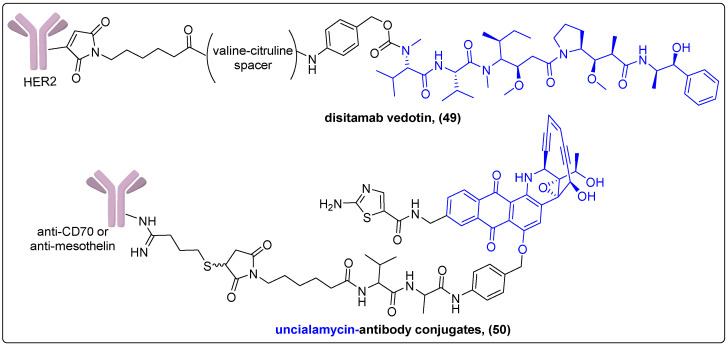
ADCs in preclinical and clinical studies.

**Figure 16 molecules-27-06632-f016:**
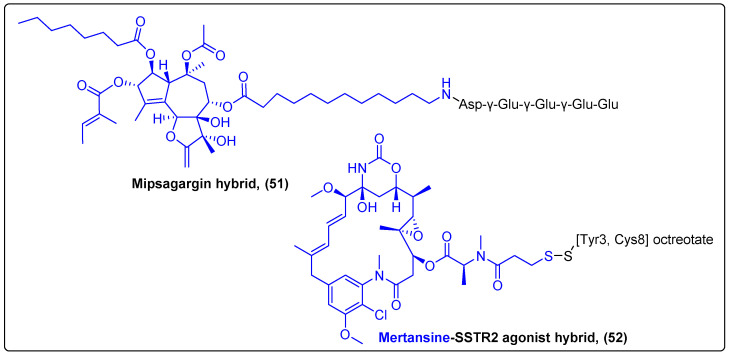
Mipsagargin and PEN-221 hybrids.

**Figure 17 molecules-27-06632-f017:**
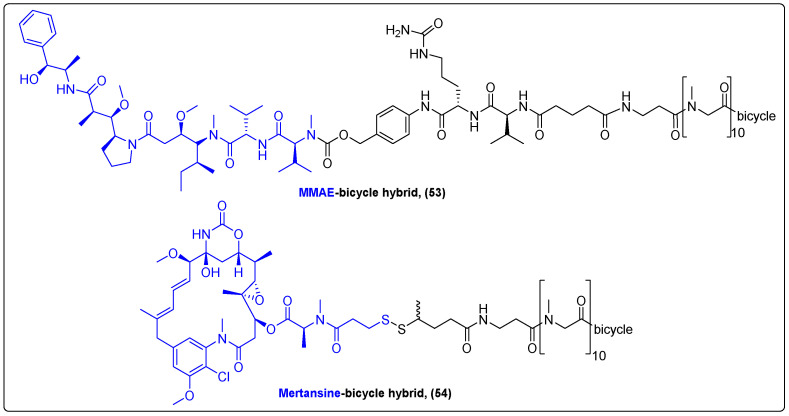
Peptide conjugates of MMAE and mertasine.

**Figure 18 molecules-27-06632-f018:**
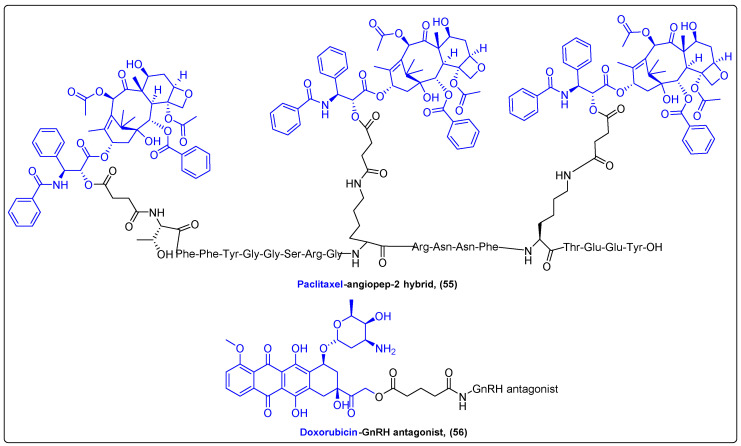
Natural product hybrids ANG1005 and Zoptrex.

**Figure 19 molecules-27-06632-f019:**
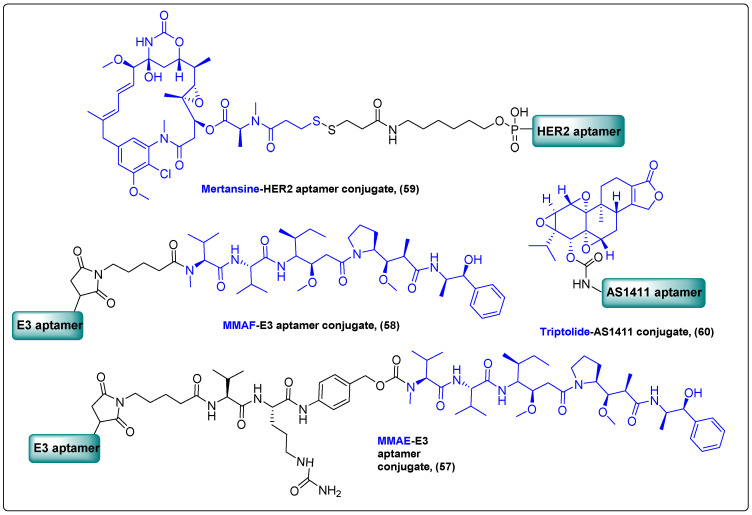
Aptamer–drug conjugates.

**Figure 20 molecules-27-06632-f020:**
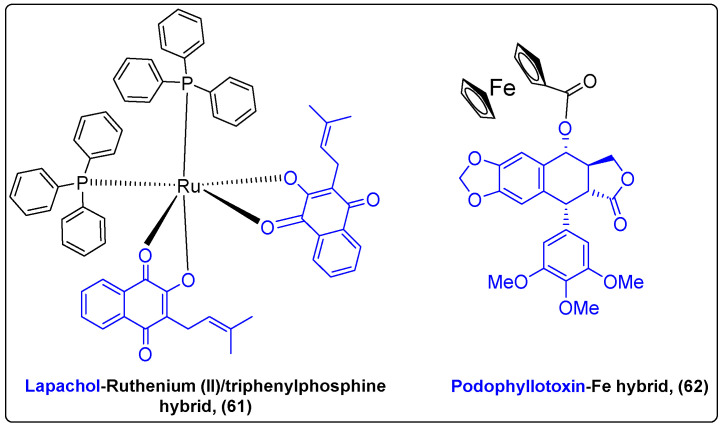
Natural product-based hybrids with metal complexes.

**Table 1 molecules-27-06632-t001:** Natural product-based hybrids commented in this work.

Natural Product	Bioactive Moiety	Conjugation	Preclinical/Clinical Studies	Ref
Thymoquinone	5-FU	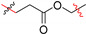	Colorectal cancer	[13]
Chalcone	1,2,3-triazole	Ether bond	Hepatocellular carcinoma	[17]
Coumarin	dihydrotriazole	Amide bond	Lung cancer	[20]
Apigenin	Piperazine	Methylene group	Lung and ovarian cancer	[23]
Acridine	Thiophene	Direct conjugation	Colorectal cancer	[26]
Quinolone	Thiazole	Direct conjugation	Phase I and IIAML, myelodysplastic syndrome	[27]
Isatin	Piperazine	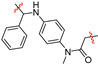	Phase II pancreatic cancer	[29]
Quinazoline	Furan	Direct	>10 active clinical trials	[30]
Pterostilbene	Vorinostat	Ether bond	Breast cancer	[39]
Isatin	Pyrrole	Direct conjugation	Phase IIthymoma, glioblastomaosteosarcoma	[28]
Resorcinol	Phenyl	Amide bond	Lung cancer	[45]
1,4-Napthoquinone	Phenyl	Direct conjugation	Melanoma	[50]
JH-VIII-49	Pomalidomide	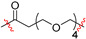	Leukemia	[54]
Wogonin	Pomalidomide	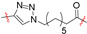	Breast cancer	[55]
Pseudolaric acid	Thalidomide	Ethylene diamine	Melanoma	[56]
Ursolic acid	Thalidomide	POE-4 linker	Lung, hepatoma	[57]
Indirubin	Pomalidomide	Diethylene glycol	Leukemia	[59]
Platanic acid	LCL-161	Diethylene glycol	Lung cancer	[61]
Nimbolide	JQ1	1,3-diaminopropane	Breast cancer	[63]
Bardoxolone	JQ1	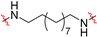	Breast cancer	[66]
Piperlongumine	SNS-032	Piperazine secondary amide	Leukemia	[69]
Fumagillin	YTK-105	PEG linker	Renal cancer	[72]
DM1	Folic acid	Disulfide bond	Lung cancer	[76]
Desacetylvinblastine monohydrazide	Folic acid	Peptide spacer	Phase I, II and IIIOvarian, small cell lung cancer	[76]
Epothilone A	Folic acid	Saccharo-peptidic spacer	Phase I/IIa solid tumors	[76]
Tubulysin	Folic acid	Saccharo-peptidic spacer	Phase I Cervical cancer	[77]
Isatin	Epiandrosterone, androsterone	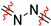	Gastric, melanomaHepatocellular carcinoma	[81,82]
Pyropheophorbide	17α-Testosterone, 17β-epitestosterone	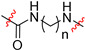	Prostate cancer	[83]
Doxorubicin	Dexamethasone	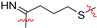	Breast cancer	[84]
Magnolol	Sulforaphane	Ether	Breast cancer	[90]
Artemisinin	Daumone	Carbamate	Breast cancer	[95]
Formonentin	Umbelliferone	1,2,3-triazole	Gastric cancer	[100]
β-Carboline	β-carboline	*N*-acyl hydrazone	Sarcoma	[104]
Mertansine	Anti-HER2	Maleimide-thiol	Approved for ErbB2-positive advanced breast cancer	[112]
MMAE	Anti-CD30	Maleimide-thiol	Approved for relapsed or refractory systemic anaplastic large-cell lymphoma	[113]
*N*-acetyl gamma-calicheamicin-dimethyl hydrazide	Anti-CD22	Hydrazone	Approved for relapsed or refractory CD22+ acute lymphoblastic leukemia	[114]
*N*-acetyl gamma-calicheamicin-dimethyl hydrazide	Anti-CD33	Hydrazone	Approved for CD 33+ acute myeloid leukemia	[115]
MMAE	Anti-CD79b	Maleimide-thiol	Approved for relapsed or refractory diffuse large B cell lymphoma	[116]
Exatecan	Anti-HER2	Maleimide-thiol	Approved for ErbB2-positive metastatic breast cancer	[118]
MMAE	Nectin-4	Maleimide-thiol	Approved for advanced or metastatic urothelial cancer	[119]
SN-38	TROP2	Maleimide-thiol	Approved for metastatic triple-negative breast cancer	[120]
MMAF	IgG1κ	Maleimide-thiol	Approved for multiple myeloma	[122]
MMAE	Anti-HER2	Valine-citruline	Phase I and IIbreast, urothelial, gastric cancer	[123]
Uncialamycin	Anti-mesothelin	Peptide	Lung cancer	[128]
Thapsigargin	Asp-γ-Glu-γ-Glu-γ-Glu-Glu	Amide bond	Phase IIadvanced refractory hepatocellular carcinoma	[131]
Mertansine	[Tyr3, Cys8] octreotate	Disulfide bond	Small-cell lung cancer	[133]
MMAE	Bicyclic peptide	Glutaric acid/peptide	Prostate cancer	[134]
Mertansine	Bicyclic peptide	Disulfide bond	Phase I/IIaadvanced solid tumors	[136]
Doxorubicin	GnRH analogue	Ester and glutaric acid	Phase IIIendometrial	[137]
Paclitaxel	Angiopep-2	Ester bond	Phase I	[138]
MMAE, MMAF	E3 aptamer	Peptide spacer/aliphatic	Prostate cancer	[140]
Mertansine	HER2 RNA aptamer	Disulfide bond	Breast cancer	[141]
Triptolide	AS1411	Amide bond	Triple-negative breast cancer	[142]
Lapachol	Ruthenium (II)/PPh_3_	Direct	Lung, breast cancer	[145]
Podophyllotoxin	Ferrocenyl moiety	Ester	Breast cancer	[146]

## Data Availability

Not applicable.

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
