# Peer review of "Recent Advances in Natural Product-Based Hybrids as Anti-Cancer Agents"

_molecules, 2022, doi:10.3390/molecules27196632_

Round 1
Reviewer 1 Report
Abstract
· Please provide a simple working definition for cancer and natural product hybrids.
· “Natural products have been used in medicine and particularly in cancer therapy for thousand years.”, I think this statement is too subjective and conclusive,especially mentioning the usage in cancer therapy for thousand years.
· The abstract is too general, I would suggest the authors to include more specific information based on the review. Include important remarks from the review in the abstract would be good.
· The aim of this review is not mentioned.
Introduction
· It is well written, but I would suggest the authors to add the aim of this review.
Line 82: Specify the plant species
Line 115: Provide a simple introductory statement of piperazine, as this is also a natural compound.
Line 121: Add paragraph spacing
Line 125-127: Rephrase this statement, it is not easy to understand.
Line 208: Specify the protein
Line 613: In 2020 (comma is missing)
I appreciate the authors on the effort to conduct a very comprehensive literature search. I would suggest authors to include summarize the findings of the review in a table form, to make the article easy to understand. Besides, many literatures are presented without any evaluation, such as which hybrid is more efficient? Which hybrid has more advantages.
Author Response
"Please see the attachment

Reviewer 2 Report
This manuscript provides a reasonably comprehensive and interesting review of recent advances in nature products-based hybrids as anti-cancer agents. Meanwhile, several nice examples of application of nature products-based hybrids are also included in this review with details. This review is well categorized and summarized, and also is very interesting and informative for the reader. This manuscript is recommended for publication after the following comments are addressed.
1. From line 67 to 69. The drug resistance issue of 5-FU should be included before this sentence, otherwise the reader will not understand why the author wanted to design 5-FU and nature products hybrids;
2. Line 99, there should be a reference after “(Figure 2)”;
3. Line 125, it is better to move the high toxicity issue to the former sentence to easy understand. For example: “…widely studied as anticancer agents against leukaemia, but the high toxicity issue of acridines that hampers its clinical usefulness. Lisboa et al…”;
4. From line 161 to 162, This sentence is uncompleted. “Activation of STAT3 limits response to HDAC inhibition” should be included in this sentence;
5. Line 167, “against” should be “in 4T1 allograft mice model”;
6. Line 173, delete “great”;
7. Line 178, “Hsp90 inhibitors pose” should “single inhibition of Hsp90 pose”
8. Line 188, “Recent literature demonstrates” should be “Recent literatures demonstrate”
9. Line 192, delete “great”;
10. Line 193, it is better to give exact IC50 values or EC50 values rather than use “mid nanomolar activity” to give more information.
11. Line 209, This sentence is not well described, and need to reorganize. Actually, the “cereblon E3 ubiquitin ligase” is not the target, CDK8 is.
12. Line 362, to give more information, it is to give the exact fold rather than use the word greater.
13. Line 364, using compound 33 instead of “The novel agent”.
14. It is unclear why only the chemical structures of compounds 43, 44 and 46 were not included in this manuscript.
15. Line 566, It is unclear the meaning of “large tumors”. Does the author mean solid tumor?
16. Line 574, BT5528 and BT1718 were not mentioned in this manuscript.
17. The references should be written following this journal’s style.
Small issues:
1. In vitro and in vivo should be written in italic.
2. Figure 3, the compound should be placed in the number order rather than randomly.
3. Line 166, “KD” should be “KD”
4. Line 193, “14” should be bold as “14”. Line 268 and 308 have the same issue.
5. Line 505, “3.4 mg.kg” should be “3.4 mg/kg”
Author Response
"Please see the attachment."

Round 2
Reviewer 2 Report
All my concerns have been well addressed, the manuscript can be published as it is.